# ReDebias: Exploring Residual energy based Debias learning

## Abstract

In real-world applications, ensuring that model decisions are independent of the training data distribution is crucial for safely deploying models. To address the long-tailed problem, massive approaches focus either on improving individual prediction quality or enhancing aggregate evaluation. Although these methods improve overall performance, they often sacrifice performance in some classes, undermining the goals of long-tailed learning. We conduct a mathematical analysis of the limitations of the Empirical Risk Minimization (ERM) framework in long-tailed learning, examining both individual performance and aggregate evaluation. For individual evaluation, although the Negative log-likelihood (NLL) metric is effective, it relies heavily on softmax leading to poor distinction and ambiguity when the probabilities of correct and incorrect predictions are similar. For aggregate evaluation, the naive estimator in ERM is not an unbiased estimator, dominated by head classes. To overcome these challenges, we propose ReDebias, a comprehensive framework combining the Residual-Energy score and a Debias estimator. The Residual-Energy score provides a more sensitive reflection of prediction quality than softmax-based scores, enhancing prediction precision and reducing ambiguity. The Debias estimator applies causal inference techniques to ensure unbiased estimates during the averaging process, correcting for class-wise biases inherent in the naive estimator. Through extensive validation on long-tailed benchmarks, including training from scratch on iNaturalist18, ImageNet-LT, and CIFAR10/100-LT, as well as fine-tuning Vision Transformer (ViT) on iNaturalist18, our method outperforms the state-of-the-art algorithms. Our code and trained models will be made available following the publication of this paper.

## 1 Introduction

In real-world scenarios, data often follow Zipf's law, resulting in long-tailed class distributions (Reed (2001)). This imbalance poses a challenge for deep neural networks trained with Empirical Risk Minimization (ERM), as they tend to favor head classes with abundant samples while neglecting tail classes with fewer samples (Zhang et al. (2023)). Such bias hinders model performance in real-world applications with long-tail distributions, such as visual question answering (Dai et al. (2023); Shi et al. (2024)), medical image diagnosis (Huang et al. (2024); Holste et al. (2024)), and unmanned aerial vehicle detection (Yu et al. (2021); Pan et al. (2023)).

To tackle this challenge, recent studies have focused on two main goals: enhancing individual prediction quality and ensuring aggregate evaluation remains unbiased by class distribution. For the first goal, mainstream paradigms aim to enhance network representation to improve individual sample performance, particularly for tail samples, measured by metric $\ell$. Common approaches include re-weighting Cui et al. (2019); Wang et al. (2024b); Peng et al. (2024), ensemble learning methods (Wang et al. (2021b); Li et al. (2022); Tao et al. (2023)), and decouple training (Kang et al. (2020); Xu et al. (2023)). For the second goal, numerous aim to ensure that aggregate evaluation is not dominated by head classes. Methods like modified sampling strategies (Chawla et al. (2002); Ren et al. (2020); Kang et al. (2020); Wu et al. (2021))and information mixing/transfering (Chou et al. (2020); Wang et al. (2024a); Gao et al. (2024); Rangwani et al. (2024)) balance the number or diversity of training samples. In addition, logit adjustment techniques (Cao et al. (2019); Menon et al. (2021); Hong et al. (2021a)) make the estimator less sensitive to class imbalance during training.

Despite overall performance gains, these approaches often at the expense of performance in certain classes, contradicting the purpose of long-tail learning. To better illustrate this trade-off, we evaluated two representative methods: LGLA(Tao et al. (2023)), which enhances individual sample performance through structural improvements, and DODA(Wang et al. (2024a)), which reduces class distribution effects on aggregate evaluation via data augmentation. We also explored a hybrid approach that combines both LGLA and DODA. As shown in Fig 1, the red curve (SA) reveals that although performance improves significantly across most classes, especially tail classes, some classes, particularly head classes, still experience performance drops, which is also reflected by the SR value. This indicates that neither method alone, nor their combination, fully balance aggregate evaluation with individual performance.

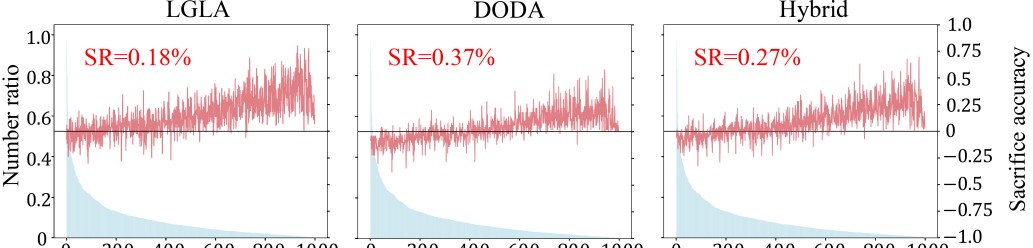

Figure 1: Comparison of class-wise performance on ImageNet-LT. Blue bars represent the distribution of class sample ratios. Red curves represent SA (Sacrificial Accuracy), indicating the difference in accuracy between baseline and Cross-Entropy (CE) for each class, with larger SA indicating greater improvement. SR (Sacrifice Ratio) quantifies the percentage of classes with performance drops compared to CE, highlighting the trade-offs in accuracy.

In response to this phenomenon, we mathematically analyze the limitations of ERM framework in long-tailed learning, focusing on both individual and aggregate evaluation tasks. For individual evaluation, we observe that Negative log-likelihood (NLL) loss, which heavily relies on softmax, struggles to distinguish correct from incorrect predictions when their predicted probabilities for the target class are similar. This weakens its ability to assess prediction quality effectively. From the perspective of energy, we find that this ambiguity arises from the misalignment between softmax-based scores and the input probability density, obscuring meaningful differences in model confidence. In aggregate evaluation, measured by estimator $\mathcal{R}$, we connect the long-tailed problem to causal inference by treating long-tailed datasets as instances of data Missing Not At Random (MNAR), where only a subset of data is observable. Under the MNAR condition, the naive estimator in ERM, which averages over observed samples, is severely biased and fails to accurately assess true performance due to the underlying data distribution bias.

To address these limitations, we introduce **Re-Debias**, a comprehensive framework combining the *Residual-Energy score* with *Debias estimator*. The *Residual-Energy score* unified predictions onto a single scale, where lower values indicate larger errors, and higher values reflect more accurate predictions. Our mathematical analysis and empirical validation confirm that this energy-based score is particularly well-suited for detection tasks, as it captures non-target class information typically missed by softmax-based scores. The *Debias estimator* corrects the inherent class-wise bias in the naive estimator of ERM by estimating and leveraging inverse propensity weights from causal inference. We theoretically prove that it is an unbiased estimator, capable of delivering more accurate performance evaluations under long-tailed distributions. To ensure Fisher consistency, we further incorporate the propensity into the logits. We validate our method through comprehensive experiments, demonstrating robust performance across various datasets and target-label distributions, confirming its effectiveness and generalizability.

In summary, the main contributions of this paper are:

- *Evaluation decomposition*: We identify that state-of-the-art ERM-based methods often achieve performance gains by sacrificing accuracy in certain classes. To address this, we decompose the optimise process into two tasks: improving individual prediction precision and ensuring unbiased aggregate evaluation.
- *Residual-Energy score*: To overcome limitations of softmax-based scores, we introduce the Residual-Energy score, an energy-based metric that captures non-target class information more accurately than softmax-based socres, improving the precision of individual predictions.

- *Debias estimator*: To correct the naive estimator, we develop a novel framework linking the long-tailed problem to causal inference, introducing the Debias estimator that ensures fair learning under long-tailed distributions through inverse propensity weighting.
- *Extensive Validation*: We validated our method by training from scratch on on iNaturalist18(Van Horn et al. (2018)), ImageNet-LT(Liu et al. (2019)), and CIFAR10/100-LT(Krizhevsky et al. (2009)), and fine-tuning Vision Transformer (ViT) (Alexey (2021)) on iNaturalist18. Our approach consistently demonstrated strong performance across different target-label distributions, confirming its effectiveness and generalization.

## 2 RELATED WORK

### 2.1 LONG-TAILED LEARNING

Long-tailed class distribution datasets are inevitable in real-world applications. Most existing long-tail classification methods solve the long-tailed problem in class-wise, and can be divided into three categories Zhang et al. (2023): class rebalancing (Menon et al. (2021); Wang et al. (2024b); Zang et al. (2021); Lin et al. (2017); Du & Wu (2023)) attempts to balance the class distribution during training; Information enhancement (Chou et al. (2020); Wang et al. (2024a; 2021a)) try to introduce additional information to improve the performance of the tail class without sacrificing the performance of the head class, for example Tang et al. (2022) utilizes the pre-trained model (Paszke et al. (2019)) to generate image feature clusters as annotations of implicit attributes, and then uses data sampling strategies to build different training environments for invariant feature learning; and module improvement Zhou et al. (2023); Jin et al. (2023); Zhou et al. (2020); Tao et al. (2023)) tries to come up with solutions from exploring methods to optimize network modules to the long-tailed problem. However, all these approaches rely on softmax-based scores to measure prediction quality, which is suboptimal. From energy perspective, the softmax-based score emphasizes reducing the target class energy and increasing that of other classes but neglects sufficiently boosting non-target class energies. This leads to similar loss values for correct and incorrect predictions, diminishing the model's ability to accurately distinguish between target and non-target classes.

### 2.2 ENERGY-BASED LEARNING

Energy-based models(EBMs) (LeCun et al. (2006); Ranzato et al. (2006; 2007)), rooted in Boltzmann machines (Ackley et al. (1985)), offer a flexible framework for both probabilistic and non-probabilistic learning. In image recognition, EBMs aim to assign as low energy as possible to images from the target class and high energies to images of other classes. Recent advancements like BiDVL(Kan et al. (2022) )proposes a bi-level optimization to enhance energy-based hidden variable models, while CLEL(Lee et al. (2023)) leverages contrastive representation learning to make EBM training faster and more efficient. In GAN training, Zhao et al. (2017) leverages energy values to improve the discriminator's performance. Despite these successes, training EBMs remains challenging due to the intractability of the normalization constant. To bypass the calculation of proper normalization, Liu et al. (2020) shows that energy scores can better distinguish in- and out-of-distribution samples compared to softmax-based methods. However, existing approaches mainly focus on the energy of target or total classes, overlooking the non-target class energy. Taking a step further, we introduce the residual-energy score, which measures the energy of non-target classes, reducing the overconfidence of softmax scores and offering greater optimization flexibility without requiring normalization. Our approach directly optimizes the energy gap between boundary samples, aligning naturally with energy-based detectors. Additionally, previous EBMs have largely ignored the long-tailed class imbalance problem. To address this, we propose an unbiased performance estimator that mitigates the imbalance using causal inference principles.

## 3 RETHINKING ERM

### 3.1 PROBLEM SETUP

Let $\mathbb{O} = (x_i, y_i)_{i=1}^N$ denote the long-tailed training dataset, where $x_i$ represents an input sample and $y_i \in \{1, \ldots, C\}$ is the corresponding class label. The total number of samples is $N = \sum_{c=1}^C n_c$,

with $n_c$ indicating the number of instances in class $c$. Following a common assumption (Zhang et al. (2023)), $\pi_c = \frac{n_c}{N}$ represents the label frequency of class $c$. The classes are sorted in decreasing order of cardinality, that is, if $i_1 < i_2$, then $n_{i_1} \geq n_{i_2}$, and it holds that $n_1 \gg n_C$. Then, the imbalance ratio, which quantifies the degree of skewness in the dataset, is calculated by $\gamma = \frac{n_1}{n_C}$. The goal of Long-tailed learning is to develop well-performing deep models from datasets characterized by long-tailed class distributions. Empirical Risk Minimization (ERM) is a widely used principle, aiming to identify a hypothesis $\hat{\theta}$ that minimizes the empirical risk $\mathcal{R}(\theta)$:

$$\hat{\theta} = \arg\min_{\theta} \mathcal{R}(\theta) = \arg\min_{\theta} \frac{1}{N} \sum_{i=1}^{n} \ell(\hat{y}_i, y_i; \theta), \tag{1}$$

This raises two critical problems: is the typical individual metric $\ell$ sufficiently precise enough to reflect the discrepancy between the prediction $\hat{y}$ and the true label $y$? And, is the estimator $\mathcal{R}$ unaffected by the class distribution?

### 3.2 Typical Individual Metric from an Energy perspective

In a discriminative neural network $f(x) : \mathbb{R}^D \to \mathbb{R}^C$, an input $x \in \mathbb{R}^D$ is mapped to $C$-dimensional logits. These logits are then transformed into categorical distribution using the softmax function:

$$p(y|x) = \frac{e^{f_y(x)}}{\sum_j^C e^{f_j(x)}}, \tag{2}$$

where $f_j(x)$ represent the logit for the $j$ class label.

According to Liu et al. (2020), the core concept of energy-based models (EBMs) is to construct a *energy* function $E(x) : \mathbb{R}^D \to \mathbb{R}$ that assigns a scalar value to each input sample $x$. Building on recent advancements highlighted by Wu et al. (2023), a set of energy values could be turned into a probability density $p(x)$ via the Boltzmann distribution:

$$p(y|x) = \frac{e^{-E(x,y)}}{\sum_{j=1}^C e^{-E(x,j)}} = \frac{e^{-E(x,y)}}{e^{-E(x)}}. \tag{3}$$

By connecting Eq.3 and Eq.2, the energy can be defined as $E(x, y) = -f_y(x)$, and the free energy function $E(x; f)$, which marginalizes over $y$, can be represented as the denominator of the softmax activation as $E(x; f) = -\log \sum_j^C e^{f_j(x)}$.

The Negative log-likelihood (NLL) loss depends on softmax, a typical individual metric $\ell$ used in ERM, is commonly defined as:

$$\ell_{nll} = -\log p(y|x) = -\log \frac{e^{f_y(x)}}{\sum_{j=1}^C e^{f_j(x)}}. \tag{4}$$

Using the energy-based formulation, this softmax-based score can be equivalently rewritten as:

$$\ell_{nll} = -\log \frac{e^{f_y(x)}}{\sum_{j=1}^C e^{f_j(x)}} = -\log e^{f_y(x)} + \log \sum_{j=1}^C e^{f_j(x)} = E(x, y) - E(x; f). \tag{5}$$

This formulation highlights that the NLL loss $\ell_{nll}$ which heavily relies on softmax, quantifies the difference between the energy $E(x, y)$ and the free energy $E(x; f)$. Minimizing the NLL loss involves decreasing $E(x, y)$ while increasing $E(x; f)$. To directly illustrate the limitations of softmax-based scores, Fig. 2 compares two samples from the CIFAR10 dataset with an imbalance ratio of 50. Even though sample 1 is correctly classified and sample 2 is misclassified, both samples yield nearly identical softmax-based score. This similarity reflects the softmax-based score tendency to saturate the output probabilities, obscuring meaningful differences in model confidence between correctly and incorrectly classified samples. Therefore, relying solely on softmax introduces ambiguity in assessing classification performance, potentially hindering model training and evaluation.

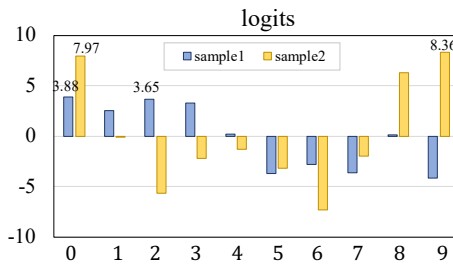 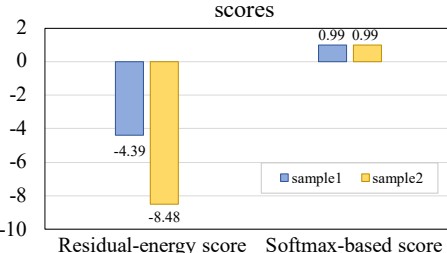

Figure 2: Comparison of two CIFAR-10-im50 samples on Resnet32. For the two samples with a target class of 0, by comparing the logits, it is clear that Sample 1 predicts correctly while Sample 2 predicts incorrectly. However, their softmax-based score (NLL loss) are nearly identical. In contrast, the residual-energy score calculated from logits of non-target classes shows a clear difference, with sample 1 at -4.39 and sample 2 at -8.48. This demonstrates that residual-energy scores provide a more sensitive reflection of prediction errors, whereas softmax-based scores do not reflect this effectively.

### 3.3 Typical stand Estimator

The naive estimator in ERM calculates empirical risk by averaging loss over the observed dataset. In an ideal scenario the class distribution without bias, and all samples are fully observed (Schnabel et al. (2016)), the standard empirical risk estimator can be expressed as:

$$R(\hat{Y}, Y) = \frac{1}{C \cdot n} \sum_{c=1}^{C} \sum_{i=1, y_i=c}^{n} \ell(\hat{y}_i, y_i), \tag{6}$$

where each class $c \in \mathcal{C}$ has $n$ instances, implying a balanced dataset with an imbalance ratio $\gamma = 1$.

However, in real-world applications, training datasets are often incomplete. Let $\mathcal{O}_{i,c} = 1$ denote that the $i$-th sample of class $c$ is observed. The conventional, naive estimator $R_{naive}(\hat{Y}, Y)$ measure the overall performance by averaging over the observed instances:

$$R_{naive}(\hat{Y}, Y) = \frac{\sum_{\{(i,c):\mathcal{O}_{i,c}=1\}} \ell(\hat{y}_i, y_i)}{|\{(i,c) : \mathcal{O}_{i,c} = 1\}|}. \tag{7}$$

A highlighted by Steck (2013), $R_{naive}(\hat{Y}, Y)$ does not provide an unbiased estimate of the true performance $R(\hat{Y}, Y)$ when the data is Missing Not At Random (MNAR):

$$\mathbb{E}_{\mathcal{O}}[\hat{R}_{naive}(\hat{Y}, Y)] \neq R(\hat{Y}, Y). \tag{8}$$

This formulation illustrates that under MNAR conditions, the naive estimator introduces bias, failing to accurately reflect the true performance of the model across the entire dataset.

## 4 Residual energy based debias estimator: an optimised estimator in ERM

To ensure that aggregate evaluation is independent of class distribution while improving the precision of individual predictions, we first introduce Residual-energy score to tackle the limination of softmax-based score (Section 4.1). Subsequently, we connect the long-tailed problem to causal inference and propose Debias Estimator which is measure the predictions quality without bias from class distribution (Section 4.2).

### 4.1 Residual-Energy-based Individual Metric

In this section, we aim to evaluate the quality of individual samples more precisely. As discussed in Section 3.2, the NLL loss, one of tipycal softmax-based scores, only consider the difference between

$E(x, y)$ and $E(x; f)$, which can lead to ambiguity in model confidence. To overcome this limitation, we propose the *Residual Energy* score:

$$E(x, \overline{y}) = -\log \sum_{j \neq 1}^{C} e^{f_j(x)}, \tag{9}$$

this score accounts for the energy of all non-target classes. As demonstrated in Fig.2, the residual energy scores offer clearer distinctions between samples, providing more meaningful information than softmax-based scores. This makes the Residual-Energy score more sensitive to variations in logit distributions, offering a finer-grained assessment of prediction certainty, especially in long-tailed scenarios where softmax scores may obscure true uncertainty.

Given the non-linear relationship between $E(x, y)$, $E(x; f)$, and $E(x, \overline{y})$, directly incorporating the residual energy $E(x, \overline{y})$ into the NLL loss is suboptimal. Therefore, inspired by Mixture of Softmax (MoS) (Yang et al. (2018)), we apply $K$ different softmax functions and mix them enhance expressive power beyond traditional softmax by residual-energy score:

$$p_{re}(y|x) = \sum_{k=1}^{K} w^k \frac{e^{f_y^k(x)}}{\sum_{j=1}^{C} e^{f_j^k(x)}}; \quad s.t. \sum_{k=1}^{K} w^k = \sum_{k=1}^{K} w(E^k(x, \overline{y})) = 1, \tag{10}$$

where $f_j^k$ is the $k$-th component associated with class $j$, $E^k(x, \overline{y})$ is the residual energy of $k$-th softmax function, and $w(\cdot)$ is a normalisation function. Thus, the NLL loss relies on residual-energy score, $\ell_{re} = -\log p_{re}(y|x)$, is able to fully consider the energy of the target class, the energy of the other classes, and the entire energy.

## 4.2 DEBIASED PERFORMANCE ESTIMATOR

Instead of evaluating the precision of individual predictions, we also want to evaluate overall performance more impartially. The key to handling the bias in naive estimator $\mathcal{R}_{naive}$ lies in a thorough understanding of the process that generates the observation instances in $\mathcal{O}$, which is known as the *Assignment Mechanism* in causal inference (Imbens & Rubin (2015)). In this paper, we assume that the assignment mechanism is probabilistic. The marginal probability of observing a sample $x_i$ with target class $y_i = c$ is denoted by $P_{i,c} = P(\mathcal{O}(x_i, y_i) = 1)$. Following Schnabel et al. (2016), observations are assumed to be independent given $P$, corresponding to a multivariate Bernoulli model where each $\mathcal{O}_{i,c}$ is a biased coin flip with probability $P_{i,c}$. In long-tailed datasets, this assignment is non-random, and $Pi, c$ varies across classes. The Inverse Propensity Scoring (IPS) estimator (Imbens & Rubin (2015)) refer to $P_{i,c}$ as the propensity of observing $(x_i, y_i)$, where $y_i = c$. IPS corrects dataset bias by inversely weighting the prediction error based on the propensity of observation, resulting in an unbiased performance estimate:

$$\hat{R}_{IPS}(\hat{Y}|P) = \frac{1}{C \cdot n} \sum_{\substack{(i,c) \in \mathcal{O}_{i,c}=1}} \frac{\ell(\hat{y}_i, y_i)}{P_{i,c}}, \tag{11}$$

where $C \cdot n$ is the total number of the ideal dataset. However, accurately estimating $P_{i,c}$ is crucial, the IPS often suffers from high variance in propensity, leading to oscillating training losses and poor generalization. Moreover, the $C \cdot n$ is often unknown in the real world. Therefore, IPS is unsuitable for directly handling the long-tail problem in deep learning.

To develop an unbiased estimator, inspired by IPS, a straightforward idea is to adjust the naive estimator for long-tailed class distributions to align with IPS:

$$\frac{1}{|\mathcal{O}|} \sum_{\mathcal{O}} \frac{\ell(\hat{y}_i, y_i)}{\widetilde{P}_{i,c}} = \frac{1}{C \cdot n} \sum_{\mathcal{O}} \frac{\ell(\hat{y}_i, y_i)}{P_{i,c}}, \tag{12}$$

Thus, the required propensity $\widetilde{P}_{i,c}$ can be reconstructed as:

$$\widetilde{P}_{i,c} = C \cdot \pi_c, \tag{13}$$

where $\pi_c$ is the label frequency, calculated as $\frac{n_c}{|\mathcal{O}|}$. The detailed process is outlined in **Appendix A.1**. As a result, we derive the Debias estimator, which inversely weights prediction quality using

$\widetilde{P}_{i,c}$ and averages over observed samples, yields an unbiased estimator. The proof is provided in **Appendix A.2**. Since the most common individual metric $\ell$ in deep learning is the NLL loss, the Debias estimator can be further expressed as:

$$R_{debias}(\hat{Y}, Y) = \frac{1}{|\mathcal{O}|} \sum_{\mathcal{O}} \frac{\ell(\hat{y}_i, y_i)}{\widetilde{P}_{i,c}} = \frac{1}{|\mathcal{O}|} \sum_{\mathcal{O}} \frac{\ell(\hat{y}_i, y_i)}{C \cdot \pi_c} = \frac{1}{|\mathcal{O}|} \sum_{\mathcal{O}} \frac{-\log(p(y_i|x_i))}{C \cdot \pi_c}. \quad (14)$$

Recalling the goal of ERM is to minimize losses across all samples:

$$\arg\min -\frac{\log(p(y_i|x_i))}{C \cdot \pi_c} = \arg\max \frac{\log(p(y_i|x_i))}{C \cdot \pi_c} \propto \arg\max \frac{p(y_i|x_i)}{C \cdot \pi_c}, \quad (15)$$

and since $p(y_i|x_i;\theta) \propto e^{f_{y_i}(x_i)}$, we have,

$$\arg\max \frac{p(y_i|x_i)}{C \cdot \pi_c} \propto \arg\max \frac{e^{f_{y_i}(x_i)}}{C \cdot \pi_c} = \arg\max(e^{f_{y_i}(x_i) - \log(C \cdot \pi_c)}). \quad (16)$$

Following Menon et al. (2021), to ensure Fisher consistency and robust network performance, we apply a class prior offset directly during logits learning, rather than post-hoc during inference. This leads to the adjusted logits $g_y(x) = f_y(x) + \log(C \cdot \pi_y)$, incorporating both marginal probabilities and the number of classes for unbiased performance evaluation:

$$\widetilde{R}_{debias}(\hat{Y}, Y) = \frac{1}{|\mathcal{O}|} \sum_{\mathcal{O}} -\log \frac{e^{g_y(x)}}{\sum_{j \in \mathcal{C}} e^{g_j(x)}} = \frac{1}{|\mathcal{O}|} \sum_{\mathcal{O}} -\log \frac{e^{f_y(x) + \log(C \cdot \pi_y)}}{\sum_{j \in \mathcal{C}} e^{f_j(x) + \log(C \cdot \pi_j)}}. \quad (17)$$

Finnally, we can combine residual-energy score with the Debias estimator to address the long-tailed problem:

$$R_{re-debias}(\hat{Y}, Y) = \frac{1}{|\mathcal{O}|} \sum_{\mathcal{O}} -\log \sum_{k=1}^{K} w^k \frac{e^{g_y^k(x)}}{\sum_{j=1}^{C} e^{g_j^k(x)}},$$

$$= \frac{1}{|\mathcal{O}|} \sum_{\mathcal{O}} -\log \sum_{k=1}^{K} w^k \frac{e^{f_y^k(x) + \log(C \cdot \pi_y)}}{\sum_{j=1}^{C} e^{f_j^k(x) + \log(C \cdot \pi_j)}}; \quad (18)$$

$$s.t. \sum_{k=1}^{K} w^k = \sum_{k=1}^{K} w(E^k(x, \overline{y})) = 1.$$

## 5 EXPERIMENTS

### 5.1 SETUP

**Long-tailed datasets** We conducted experiments on four long-tailed datasets: CIFAR-10-LT, CIFAR-100-LT, ImageNet-LT, and iNaturalist18. CIFAR-10-LT and CIFAR-100-LT have imbalance ratios of 50 and 100. ImageNet-LT is derived from the larger ImageNet dataset, and it consists of 1,000 classes with images ranging from 1,280 to 5 per class, while iNaturalist 2018 showcases a naturally long-tailed distribution with samples from over 8,000 species.

**Training Details** For a fair comparison, we follow the setup outlined in previous works(Menon et al. (2021)). Unless specified, we use an SGD optimizer with momentum 0.9 and a weight decay of $10^{-4}$. Cosine learning rate decay and standard data augmentation are also applied as in prior works(Menon et al. (2021)).

In the *traning from scratch* experiments, for CIFAR-10/100-LT, we train ResNet-32 from scratch for 200 epochs with a batch size of 128. The base learning rate is set to 0.4, with a 5-epochs linear warm-up, followed by decay factors of 0.1 at the 160th and 180th epochs. For ImageNet-LT, we train ResNext-50 from scratch, using a base learning rate of 0.05, and a batch size of 256, with a weight decay of $5 \times 10^{-4}$, as in Tao et al. (2023). For iNaturalist 2018, we adopt ResNet-50 with a base learning rate of 0.1, a batch size of 512, and a weight decay of $2 \times 10^{-4}$.

In the *fine-tuning* experiments, we employ ViT-B/32(Dosovitskiy et al. (2020)) as the backbone, performing end-to-end fine-tuning for 100 epochs with a base learning rate is 0.01 and a weight decay of $2 \times 10^{-4}$.

**Evaluation Metrics** We present our evaluation results using top-1 accuracy, denoted as "All". In accordance with Kang et al. (2020), we further categorize the validation/test sets of ImageNet-LT and iNaturalist18 into three subsets based on the number of training instances: Many (more than 100 instances), Medium (20 to 100 instances), and Few (less than 20 instances). This breakdown allows for a more nuanced analysis of performance across varying sample sizes in the long-tailed distribution.

## 5.2 RESULTS ON TRAINING FROM SCRATCH

We compare our method with various baselines that address long-tailed problems, categorize them into three main approaches, as Zhang et al. (2023): Class Re-balancing, Module Imporvement, and Information Augmentation. The detailed descriptions for baselines are in **Appendix A.3**

Table 1: Breakdown results of *training from scratch* on ImageNet-LT and iNaturalist18. The epo is an abbreviation for epochs, and Med is short for Medium. "*": results reported in OTmix. "†": results reported in origin paper. RIDE and RIDE-based methods have 3 experts by default. The best and second best performance for each dataset configuration are **bolded** and underlined, respectively.

| Methods | ImageNet-LT | | | | | iNaturalist18 | | | | |
|---|---|---|---|---|---|---|---|---|---|---|
| | epo | Many | Med | Few | All | epo | Many | Med | Few | All |
| ViT-B Backbone training from scratch | | | | | | | | | | |
| ViT† | 800 | 56.9 | 30.4 | 10.3 | 37.5 | 800 | 64.3 | 53.9 | 52.1 | 54.2 |
| DeiT† | 800 | 70.4 | 40.9 | 12.8 | 48.4 | 800 | 72.9 | 62.8 | 55.8 | 61.0 |
| LiVT† | 900 | 73.6 | 56.4 | 41.0 | 60.9 | 900 | 78.9 | 76.5 | 74.8 | 76.1 |
| DeiT-LT† | 1400 | 66.6 | 58.3 | 40.0 | 59.1 | 1000 | 70.3 | 75.2 | 76.2 | 75.1 |
| ResNext-50 Backbone training from scratch | | | | | ResNet-50 Backbone training from scratch | | | | | |
| CE * | 90 | 66.8 | 38.4 | 8.4 | 45.3 | 200 | 73.9 | 63.5 | 55.5 | 61.0 |
| Focal loss* | 90 | 66.9 | 39.2 | 9.2 | 45.8 | 200 | - | - | - | 61.1 |
| Logits Adj.† | 90 | 62.2 | 49 | 28.3 | 51.3 | 90 | - | - | - | 68.4 |
| LADE† | 180 | 65.1 | 48.9 | 33.4 | 53.0 | 200 | - | - | - | 70.0 |
| BALMS* | 90 | 50.3 | 39.5 | 25.3 | 41.8 | 200 | 70.0 | 70.2 | 69.9 | 70.0 |
| DDC† | 400 | 62.9 | 52.6 | 37.1 | 54.1 | 400 | 64.7 | 70.7 | 72.1 | 70.7 |
| PaCo† | 400 | 67.2 | 56.9 | 36.7 | 58.2 | 400 | - | - | - | 73.2 |
| TDE† | 90 | 62.7 | 48.8 | 31.6 | 51.8 | 200 | - | - | - | 63.9 |
| BBN† | 90 | 52.6 | 46.3 | 43.8 | 49.3 | 200 | 49.4 | 70.8 | 65.3 | 66.3 |
| Decouple-cRT* | 100 | 58.8 | 44.0 | 26.1 | 47.3 | 200 | 69.0 | 66.0 | 63.2 | 65.2 |
| Decouple-LWS* | 100 | 57.1 | 45.2 | 29.3 | 47.7 | 200 | 65 | 66.3 | 65.5 | 65.9 |
| RIDE † | 100 | 66.2 | 51.7 | 34.9 | 54.9 | 100 | 70.9 | 72.4 | 73.1 | 72.6 |
| NCL† | 200 | - | - | - | 60.5 | 400 | 72.7 | 75.6 | 74.5 | 74.9 |
| LGLA† | 180 | - | - | - | 61.1 | 400 | 70.1 | 76.2 | 77.6 | 76.2 |
| Remixup* | 100 | 60.4 | 46.9 | 30.7 | 48.6 | 200 | - | - | - | 62.3 |
| CMO+CE* | 100 | 67 | 42.3 | 20.5 | 49.1 | 200 | 76.9 | 69.3 | 66.6 | 68.9 |
| CMO+RIDE* | 100 | 66.4 | 53.9 | 35.6 | 56.2 | 200 | 68.7 | 72.6 | 73.1 | 72.8 |
| OTmix+CE† | 200 | 70.0 | 45.9 | 22.3 | 52.0 | 210 | 69.3 | 70.5 | 68.4 | 69.5 |
| OTmix+RIDE† | 200 | 59.4 | 56.5 | 44.1 | 57.3 | 210 | 71.3 | 72.8 | 73.8 | 73.0 |
| DODA+CE † | 100 | 67.4 | 47.5 | 13.9 | 48.1 | 100 | 74.9 | 66.0 | 58.4 | 63.6 |
| DODA+RIDE † | 100 | 66.9 | 54.1 | 37.4 | 56.9 | 100 | 71.2 | 73.2 | 73.4 | 73.7 |
| ours | 90 | 72.8 | **61.3** | 42.1 | 63.1 | 90 | 76.7 | 75.8 | 76.0 | 76.0 |
| ours | 180 | **74.6** | 60.3 | 42.7 | 63.4 | 200 | 78.8 | 78.3 | 78.2 | 78.3 |
| ours | 200 | 74.4 | 61.0 | **44.5** | **63.9** | 400 | **80.6** | **79.6** | **79.1** | **79.5** |

To demonstrate the scalability and effectiveness of our method, we evaluated it on two large-scale, real-world long-tailed datasets: ImageNet-LT and iNaturalist18. For fair comparison, we report results at 90, 180, and 200 epochs for ImageNet-LT, and at 90, 200, and 400 epochs for iNaturalist18. As shown in Table 1, our method achieves a top-1 accuracy of 63.9% at 200 epochs on ImageNet-LT, with performance improving as training progresses. Notably, even after just 90 epochs, it surpasses

previous state-of-the-art results, demonstrating effectiveness early in training. This consistent improvement underscores the robustness and efficiency of our approach, which continues to outperform existing methods with extended training. Since iNaturalist18 lacks a validation set, we report test accuracy directly. Our method achieves an overall accuracy of 79.5% at 400 epochs, outperforming the best transformer-based method, DeiT-LT (75.1%), and RIDE (72.6%). It also maintains strong performance across Many (80.6%), Medium (79.6%), and Few (79.1%) categories, highlighting its robustness in handling long-tailed distributions.

Table 2 shows that our method outperforms all baselines across various imbalance ratios for both CIFAR-10-LT and CIFAR-100-LT datasets. Specifically, our method achieves the highest accuracy of 88.31% and 91.51% for CIFAR-10-LT with imbalance ratios of 100 and 50, respectively. In the CIFAR-100-LT dataset with an imbalance ratio of 100, our method performs slightly worse than LGLATao et al. (2023). This discrepancy may be due to underfitting, as our model is trained for only 200 epochs, while LGLA was trained for 400 epochs.

Table 2: Top 1 accuracy for CIFAR-10/100-LT. "*": results reported in LGLA. "†": results reported in origin paper. The best and second-best performances are **bolded** and underlined, respectively.

| Methods | CIFAR-10-LT | | CIFAR-100-LT | |
|---|---|---|---|---|
| | 100 | 50 | 100 | 50 |
| CB Focal loss* | 74.6 | 79.3 | 38.7 | 46.2 |
| Logits Adj.* | - | 77.7 | - | 43.9 |
| LADE* | - | - | 45.4 | 50.5 |
| BALMS† | 84.9 | 88.9 | 50.8 | 54.1 |
| DDC† | 83.6 | 82.3 | 46.4 | 57.9 |
| PaCo* | - | - | 52.0 | 56.0 |
| BBN* | 79.8 | 82.2 | 39.4 | 47.0 |
| RIDE* | 81.6 | 84.0 | 48.6 | 49.1 |
| TDE* | 80.6 | 83.6 | 44.1 | 50.3 |
| NCL* | 85.5 | 87.3 | 54.2 | 58.2 |
| LGLA† | 87.5 | 89.8 | **57.2** | 61.6 |
| Remixup† | 75.4 | 79.8 | 39.5 | 45.0 |
| CMO+ERM† | 75.0 | 81.4 | 43.9 | 47.3 |
| CMO+RIDE† | 82.2 | 84.6 | 50.0 | 53.0 |
| OTmix+ERM† | 78.2 | 83.4 | 46.4 | 50.7 |
| OTmix+RIDE† | 82.7 | 85.2 | 50.7 | 53.8 |
| ours | **88.3** | **91.5** | 56.3 | **62.2** |

### 5.3 RESULTS ON FINE-TUNING VIT

Efficient fine-tuning helps the model retain the general knowledge learned from a large, diverse dataset while refining its parameters to perform well on the specific task, improving accuracy and efficiency compared to training from scratch. To assess the effectiveness and scalability of our approach, we conducted end-to-end fine-tuning of the ViT model for 100 epochs on the iNaturalist18 dataset, as shown in Table 3. For comparison, we established baseline methods, with LPT(Dong et al. (2023)) and LTGC(Zhao et al. (2024)) representing prompt tuning, and VL-LTR(Tian et al. (2022)), RAC(Long et al. (2022)) and CLIP-Finetune representing end-to-end fine-tuning. Our method achieved the highest overall accuracy of 83.9%, outperforming all baselines. It demonstrated robust performance, with 84.2% on Many, 84.0% on Medium, and 83.6% on Few.

### 5.4 EVALUATION AND ANALYSIS

Table 3: Breakdown results of *fine-tuning* on iNaturalist18. "†": results reported in origin paper.

| Method | Many | Medium | Few | All |
|---|---|---|---|---|
| CLIP Zero | 6.1 | 3.3 | 2.9 | 3.4 |
| CLIP Finetune | 76.6 | 74.1 | 70.2 | 72.6 |
| VL-LTR † | - | - | - | 76.8 |
| RAC† | 75.9 | 80.5 | 81.0 | 80.2 |
| LPT† | - | - | 79.3 | 76.1 |
| LTGC† | 77.5 | 83.9 | 82.6 | 82.5 |
| ours | **84.2** | **84.0** | **83.9** | **84.0** |

In this section, we conduct a detailed analysis of the mechanism of Re-Debias and discuss the following three concerns.

**How does each component affect performance?**

We conducted an ablation study on ImageNet-LT using ResNeXt-50 as the backbone, training all models for 90 epochs. The results are presented in Table 4, where the numbers in parentheses indicate the number of softmax operations used. Applying the debias estimator significantly improves overall accuracy from 43.5% to 52.2%. Introducing the original MoS with three softmax components into the baseline increases accuracy to 47.0%, and integrating MoS with residual-energy scoring further enhances it to 52.1%. Finally, combining residual-energy-based MoS with the debias estimator yields an overall accuracy of 63.1%, with the energy-based MoS also utilizing three softmax components. These results demonstrate

that our method does not require a gating mechanism to adjust the contribution of each softmax component, as is necessary in MoS, thereby avoiding potential overfitting and added complexity.

Table 4: Ablation results on ImageNet-LT. These models are all trained on ResNext-50 by 90 epoches. The number in brackets indicates the number of softmax.

| MoS | Energy | Debias | ImageNet-LT | | | |
|-----|--------|--------|------|--------|-----|-----|
| | | | Many | Medium | Few | All |
| | | | 66.8 | 38.4 | 8.4 | 45.3 |
| ✓(3) | | | 68.2 | 40.4 | 9.9 | 47.0 |
| | | ✓ | 63 | 49.7 | 30.4 | 52.2 |
| ✓(3) | | ✓ | 62.9 | 49.4 | 31.0 | 52.1 |
| ✓(3) | ✓ | | 74.0 | 50.8 | 17.5 | 55.4 |
| ✓(2) | ✓ | ✓ | 69.1 | 55.9 | 36.7 | 58.4 |
| ✓(3) | ✓ | ✓ | 72.8 | **61.3** | **42.1** | **63.1** |
| ✓(4) | ✓ | ✓ | **74.1** | 58.8 | 38.4 | 61.9 |

We further explored the effect of varying the number of softmax components: using two components achieves an accuracy of $58.4\%$, while three components reach the highest accuracy of $63.1\%$. However, increasing to four components slightly decreases accuracy to $61.9\%$. This suggests that optimal performance is achieved with three softmax components. Our final evaluation, incorporating both the debias estimator and residual-energy scoring, highlights the importance of balancing model complexity to optimize performance in long-tailed visual recognition tasks. Additional ablation studies are provided in **Appendix A.4**.

**Does ReDebias make fewer classes be 'sacrificed'?** we analyse and validate that previous approaches improve the average performance by sacrificing certain classes (especially the header class).The goal of Re-Debias is to mitigate the imbalance in terms of data while aiming for inter-class fairness. Ambiguity is reduced by allowing residual-energy scores to enhance the expressiveness of softmax-based scores. AIn addition, the Debias estimator ensures that the class distribution no longer affects the averaging operation. From the visualization of the sacrifice accuracy of each class and sacrifice rate in Fig.3, it can be found that compared with Re-Debias, Re-Deibas reduces the sacrifice rate by $17.2\%$, indicating that Re-Debias makes fewer classes be 'sacrifaiced'.

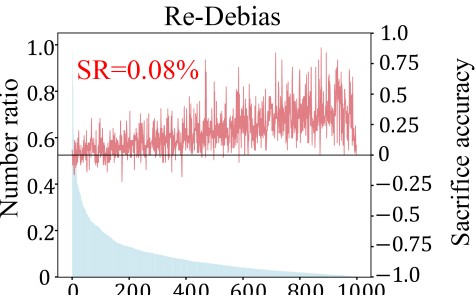

Figure 3: Visualization of the sacrifice accuracy of each class between CE and ReDebias.

**Does ReDebias still suffer from class imbalance from training?** The results in Table 5 reveal significant performance differences among methods under varying imbalance ratios (IB) during inference. In the forward label distribution scenario, while accuracies generally decrease as IB values drop, our model maintains high accuracy rates of $65.7\%$ and $65.3\%$ at IB values of 25 and 10, respectively, demonstrating robustness to different imbalance levels. Under both uniform and backward target label distributions, our method consistently performs well, achieving the highest accuracy of $61.2\%$ in the uniform setting and outperforming other methods in backward scenarios. In contrast, although methods like LADE and LGLA excel under certain conditions, they do not surpass our model overall. This indicates that our approach not only adapts effectively to forward label distributions but also maintains stable, high performance under uniform and backward distributions, highlighting its general adaptability to various target label distribution offsets.

Table 5: Top-1 accuracy over all classes on test time shifted ImageNet-LT. All models are all trained on ResNet-50. "*": results reported in LADE. IB denotes the imbalance ratio, epo represents the epoch, and Unif stands for Uniform.

| Dataset | epo | Forward | | | | | Unif | Backward | | | | |
|---------|-----|------|------|------|------|------|------|------|------|------|------|------|
| IB | - | 50 | 25 | 10 | 5 | 2 | 1 | 2 | 5 | 10 | 25 | 50 |
| CE * | 180 | 66.3 | 63.9 | 60.4 | 57.1 | 52.3 | 48.2 | 44.2 | 38.9 | 35.0 | 30.5 | 27.9 |
| TDE* | 180 | 64.1 | 62.5 | 60.1 | 57.8 | 54.6 | 52.0 | 49.3 | 45.8 | 43.4 | 40.4 | 38.4 |
| LADE* | 180 | **67.4** | 64.8 | 61.3 | 58.6 | 55.2 | 53.0 | 51.2 | 49.8 | 49.2 | 49.3 | 50.0 |
| LGLA | 180 | 64.0 | 62.7 | 61.9 | 61.0 | 59.6 | 59.9 | 57.6 | 56.9 | 56.7 | 55.9 | 56.5 |
| OTmix | 200 | 65.7 | 64.4 | 63.9 | 62.1 | 61.5 | 60.4 | 59.3 | 58.6 | 58.0 | 56.8 | 57.2 |
| ours | 90 | 66.8 | **65.7** | **65.3** | **63.9** | **62.4** | **61.2** | **60.2** | **59.3** | **58.7** | **57.8** | **58.2** |

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
