# ReDebias: Exploring Residual energy based Debias learning

## A    Appendix

### A.1    The propensity in debias estimator

To develop an unbiased estimator, a straightforward idea is to adjust the naive estimator of ERM to align with IPS:

$$
\frac{1}{|\mathcal{O}|} \sum_{\mathcal{O}} \frac{\delta(\hat{y}_i, y_i)}{\widetilde{P}_{i,c}} = \frac{1}{C \cdot N} \sum_{\mathcal{O}} \frac{\delta(\hat{y}_i, y_i)}{P_{i,c}},
$$

$$
\sum_{\mathcal{O}} \frac{\delta(\hat{y}_i, y_i)}{\widetilde{P}_{i,c}} = \frac{|\mathcal{O}|}{C \cdot N} \sum_{\mathcal{O}} \frac{\delta(\hat{y}_i, y_i)}{P_{i,c}}, \tag{1}
$$

$$
\sum_{\mathcal{O}} \frac{\delta(\hat{y}_i, y_i)}{\widetilde{P}_{i,c}} = \sum_{\mathcal{O}} \frac{\delta(\hat{y}_i, y_i)}{C \cdot N \cdot \frac{1}{|\mathcal{O}|} \cdot P_{i,c}},
$$

where $\mathcal{O}$ is short for $\mathcal{O}_{i,c} = 1$, standing for the observed sample $x_i$ with target class $y_i = c$. Thus:

$$
\widetilde{P}_{i,c} = \frac{C \cdot N \cdot P_{i,c}}{|\mathcal{O}|} = \frac{C \cdot N_c}{|\mathcal{O}|} = C \cdot \pi_c. \tag{2}
$$

### A.2    the prove

We use the propensity to adjust the naive estimator of ERM. This adjustment allows for unbiased estimates using statistical information from the observed dataset. Here, we prove that the Debias estimator is unbiased:

$$
\begin{aligned}
\mathbb{E}_{\mathcal{O}}[R_{debias}(\hat{Y}, Y)] &= \mathbb{E}_{\mathcal{O}_{i,c}}\Big[\frac{1}{|\mathcal{O}|} \sum_{\mathcal{O}} \frac{\delta(\hat{y}_i, y_i)}{\widetilde{P}_{i,c}}\Big] \\
&= \mathbb{E}_{\mathcal{O}_{i,c}}\Big[\sum_{\mathcal{O}} \frac{\delta(\hat{y}_i, y_i)}{|\mathcal{O}| \cdot \widetilde{P}_{i,c}}\Big] \\
&= \mathbb{E}_{\mathcal{O}_{i,c}}\Big[\sum_{\mathcal{O}} \frac{\delta(\hat{y}_i, y_i)}{C \cdot N \cdot P_{i,c}}\Big] \\
&= \frac{1}{C \cdot N} \mathbb{E}_{\mathcal{O}_{i,c}}\Big[\sum_{(i,c) \in \mathcal{O}_{i,c}=1} \frac{\delta(\hat{y}_i, y_i)}{P_{i,c}}\Big] \\
&= \frac{1}{C \cdot N} \sum_{c} \sum_{i=1, y_i=c} \mathbb{E}_{\mathcal{O}_{i,c}}\Big[\frac{\delta(\hat{y}_i, y_i)\mathcal{O}_{i,c}}{P_{i,c}}\Big] \\
&= \frac{1}{C \cdot N} \sum_{c} \sum_{i=1, y_i=c} \delta(\hat{y}_i, y_i) \\
&= R(\hat{Y}, Y).
\end{aligned} \tag{3}
$$

## A.3 BASELINES

We categorize the baselines into three main approaches, as Zhang et al. (2023): Class re-balancing methods, like Focal loss(Cui et al. (2019)), BALMS(Ren et al. (2020)), Logits Adj.(Menon et al. (2021)), LADE(Hong et al. (2021)), and DDC(Wang et al. (2024b)), adjust the learning process to prioritize underrepresented classes. Module improvement methods, including PaCo(Cui et al. (2021)), TDE(Tang et al. (2020)), BBN(Zhou et al. (2020)), Decouple(Kang et al. (2020)), RIDE(Wang et al. (2021)), NCL(Li et al. (2022)) and LGLA(Tao et al. (2023)), aim to enhance network representation by architectures enhance or training strategies for long-tailed learning. Information augmentation methods, such as Remixup(Chou et al. (2020)), CMO(Park et al. (2022)), OTmix(Gao et al. (2024)) and DODA(Wang et al. (2024a)), seek to introduce additional information into model training. We also compare with ViT(Dosovitskiy et al. (2020)), DeiT(Touvron et al. (2022)), LiVT(Xu et al. (2023)), DeiT-LT(Rangwani et al. (2024)), where ViT-B is the backbone, trained from scratch.

## A.4 HYPER-PARAMETER SENSITIVITY ANALYSIS OF $K$

Table 1: The number of softmax sensitivity analysis on ImageNet-LT. $K$ is the number of softmax.

| Backbone | K | 90 Epochs | | | | 200 Epochs | | | |
|---|---|---|---|---|---|---|---|---|---|
| | | Many | Med | Few | All | Many | Med | Few | All |
| ResNet-50 | 1 | 62.7 | 47.8 | 25.1 | 50.4 | 64.4 | 47.6 | 26.9 | 51.2 |
| | 2 | 68.2 | 52.3 | 31.5 | 55.6 | 69.2 | 52.9 | 34.3 | 56.6 |
| | 3 | 72.1 | 57.8 | 36.8 | 61.2 | 74.3 | 57.8 | 38.5 | 61.6 |
| | 4 | 73.1 | 57.8 | 39.2 | 61.1 | 74.5 | 58.0 | 38.9 | 61.8 |
| | 5 | 72.5 | 55.1 | 36.6 | 59.3 | 72.4 | 56.6 | 36.6 | 60.0 |
| | 6 | 73.8 | 57.0 | 38.4 | 60.9 | 74.1 | 58.3 | 38.5 | 61.7 |
| ResNext-50 | 1 | 63.0 | 49.7 | 30.4 | 52.2 | 67.8 | 51.1 | 31.4 | 54.9 |
| | 2 | 69.1 | 55.9 | 36.7 | 58.4 | 71.3 | 56.3 | 35.5 | 59.2 |
| | 3 | 72.8 | 61.3 | 42.1 | 63.1 | 74.4 | 61.0 | 44.5 | 63.9 |
| | 4 | 74.1 | 58.8 | 38.4 | 61.9 | 75.6 | 58.9 | 39.3 | 62.7 |
| | 5 | 75.1 | 58.4 | 39.3 | 62.2 | 76.4 | 58.8 | 40.1 | 63.0 |
| | 6 | 76.0 | 58.8 | 40.4 | 62.9 | 76.8 | 59.2 | 41.3 | 63.6 |
| ResNet-101 | 1 | 69.6 | 54.0 | 33.6 | 57.2 | 70.9 | 54.5 | 33.8 | 58.0 |
| | 2 | 71.0 | 55.5 | 35.3 | 58.8 | 73.5 | 56.7 | 38.1 | 60.7 |
| | 3 | 77.7 | 60.1 | 40.5 | 64.2 | 78.9 | 59.4 | 46.1 | 66.1 |
| | 4 | 77.4 | 59.2 | 41.8 | 63.8 | 78.7 | 58.9 | 47.9 | 65.4 |
| | 5 | 77.5 | 59.7 | 41.3 | 64.0 | 78.6 | 59.8 | 45.5 | 65.1 |
| | 6 | 75.0 | 58.0 | 39.3 | 62.0 | 76.4 | 59.2 | 40.6 | 63.3 |

Table 2: The number of softmax sensitivity analysis on iNaturalist18. $K$ is the number of softmax.

| Backbone | K | 100 Epochs | | | | 200 Epochs | | | |
|---|---|---|---|---|---|---|---|---|---|
| | | Many | Med | Few | All | Many | Med | Few | All |
| ResNet-50 | 1 | 69.5 | 69.2 | 69.2 | 69.2 | 70.6 | 70.6 | 70.9 | 70.7 |
| | 2 | 72.9 | 72.6 | 72.8 | 72.7 | 74.9 | 74.8 | 74.3 | 74.6 |
| | 3 | 77.9 | 77.2 | 76.8 | 77.1 | 78.8 | 78.3 | 78.2 | 78.3 |
| | 4 | 76.0 | 75.6 | 74.8 | 75.4 | 78.4 | 77.8 | 77.5 | 77.7 |
| | 5 | 73.6 | 73.8 | 72.9 | 73.4 | 75.9 | 75.1 | 74.9 | 75.1 |
| | 6 | 76.0 | 75.6 | 74.8 | 75.3 | 76.4 | 77.0 | 76.4 | 76.7 |
| ResNet-101 | 1 | 78.1 | 73.3 | 67.9 | 71.7 | 76.6 | 73.7 | 70.2 | 72.6 |
| | 2 | 76.8 | 75.5 | 73.2 | 74.7 | 78.7 | 75.9 | 74.3 | 75.5 |
| | 3 | 78.4 | 78.6 | 77.8 | 78.2 | 79.3 | 79.1 | 78.6 | 78.9 |
| | 4 | 78.2 | 79.2 | 78.7 | 78.9 | 79.8 | 80.7 | 80.5 | 80.5 |
| | 5 | 76.6 | 77.7 | 77.0 | 77.3 | 76.8 | 81.6 | 83.5 | 78.3 |
| | 6 | 77.6 | 76.9 | 75.5 | 76.4 | 78.1 | 77.3 | 76.3 | 77.0 |
| ResNet-152 | 1 | 74.8 | 71.2 | 64.4 | 73.0 | 79.5 | 76.3 | 70.8 | 74.6 |
| | 2 | 78.8 | 76.9 | 73.5 | 75.7 | 76.4 | 77.4 | 78.1 | 78.0 |
| | 3 | 80.2 | 78.8 | 77.4 | 78.4 | 80.0 | 79.6 | 78.3 | 79.1 |
| | 4 | 78.3 | 82.4 | 83.8 | 79.6 | 80.4 | 80.8 | 79.9 | 80.4 |
| | 5 | 82.0 | 80.7 | 80.6 | 80.8 | 80.9 | 81.7 | 81.4 | 81.5 |
| | 6 | 77.7 | 75.4 | 72.3 | 77.4 | 77.5 | 80.8 | 81.3 | 78.5 |
| ViT-B/32 | 1 | 77.6 | 74.4 | 70.9 | 73.3 | - | - | - | - |
| | 2 | 77.6 | 76.9 | 75.5 | 76.4 | - | - | - | - |
| | 3 | 81.1 | 81.1 | 80.9 | 81.0 | - | - | - | - |
| | 4 | 83.1 | 82.6 | 82.9 | 82.8 | - | - | - | - |
| | 5 | 84.2 | 84.0 | 83.9 | 84.0 | - | - | - | - |
| | 6 | 83.6 | 83.7 | 83.3 | 83.5 | - | - | - | - |