# OpenReview forum: "ReDebias: Exploring Residual energy based Debias learning"
_ICLR.cc/2025/Conference — ICLR 2025 Conference Withdrawn Submission_

### Official Review · Reviewer_eonk · 2024-10-19

**Soundness:** 3
**Presentation:** 2
**Contribution:** 2
**Rating:** 3
**Confidence:** 3

**Summary:**

This manuscript focuses on the long-tailed image classification task. The authors are motivated by the observed performance degradation on certain classes exhibited by previous state-of-the-art methods. To address this issue, they provide a theoretical analysis that elucidates the underlying reasons for this phenomenon from two perspectives: individual and aggregate.
Building on this analysis, the authors propose a framework named ReDebias, which seeks to mitigate the causal influences introduced by long-tailed data by surrogate modeling of the softmax scores.
To validate the effectiveness of their proposed method, the authors conduct extensive experiments across four long-tailed image classification datasets, utilizing various backbone models. These experiments are designed to demonstrate the robustness and efficacy of ReDebias in overcoming the challenges posed by long-tailed distributions.

**Strengths:**

Structure and Motivation: The manuscript presents a well-structured approach, clearly outlining the motivation behind the research and the strategies employed to tackle the identified challenges. While I may not fully agree with some aspects of the authors' claims, the overall organization effectively conveys the authors' objectives and rationale.

 Extensive Experiments: The experiments conducted in this manuscript are comprehensive, showcasing a wide range of comparisons with prior works across several mainstream benchmarks. This thorough evaluation provides valuable insights into the performance of the proposed method and strengthens the manuscript's credibility.

**Weaknesses:**

1. The writing in this manuscript presents significant challenges, particularly for readers who are not well-versed in the nuances of long-tailed distribution learning. For instance, in line L079-P2, the authors state, “we observe that Negative log-likelihood (NLL) loss, which heavily relies on softmax, struggles to distinguish correct from incorrect predictions when their predicted probabilities for the target class are similar.” However, this assertion leaves me perplexed. What exactly do the authors mean by this? Additionally, why does this occur? If such claims are central to the manuscript's argument, providing empirical evidence to substantiate them would greatly enhance the reader's understanding.

2. Similarly, in line L085-P2, the authors discuss connecting the long-tailed problem to causal inference by framing long-tailed datasets as instances of data that are Missing Not At Random (MNAR). However, the acronym MNAR is not sufficiently explained. While I interpret it as indicating that the missing data follows a certain distribution, the authors' articulation of MNAR as “only a subset of data is observable” is not clear and may lead to further confusion.

3. Another point of confusion arises when comparing the Residual-Energy score to established model calibration techniques. The authors fail to clearly articulate the distinct role of this score within the context of model calibration, which typically aims to align prediction confidence with posterior error rates. In line L090-P2, the authors describe the Residual-Energy score by stating, “where lower values indicate larger errors, and higher values reflect more accurate predictions.” However, this description does not effectively clarify how the Residual-Energy score differs from traditional model calibration methods or its unique contributions.

4. The manuscript’s motivation also raises questions about its significance. The authors suggest that previous methods, such as LGLA and DODA, demonstrate suboptimal performance in head classes, as illustrated in Figure 1. However, the differences in performance appear minimal, given that the Sacrificial Accuracy axis ranges from -1% to 1%, which may be seen as trivial. If my interpretation is correct, this diminishes the overall persuasive power of the motivation for the research.

5. Furthermore, any marginal improvement over previous works—such as the roughly 1% enhancement shown in Table 2—further reduces the perceived significance of this contribution. When considered alongside the limited advancements in performance metrics, the paper fails to convincingly establish itself as a significant advancement in the field.

6. Lack of conclusion section.

In summary, I find that the manuscript does not effectively convey its novelty or importance, primarily due to unclear writing and insufficient explanations of key concepts. The performance results also do not indicate a revolutionary contribution. As such, I recommend rejection of this submission.

**Questions:**

In addition to the questions raised in the WEAKNESSES section, I would appreciate further clarification on the following points:

 Computational Process of SR in Figure 1: Could the authors provide a more detailed explanation of the computing process for the SR (Segregation Rate) depicted in Figure 1? A step-by-step breakdown of how the SR is calculated would enhance the reader's understanding and allow for a clearer interpretation of the results presented.

 Addressing Evaluation Challenges: I would also like to request additional explanations regarding how the proposed method specifically addresses or alleviates the challenges associated with both individual and aggregate evaluation. What mechanisms or strategies does the proposed approach employ to improve performance in these contexts? Providing concrete examples or theoretical backing would strengthen the argument and clarify the significance of the proposed contributions.

Thank you for considering these requests for elaboration. They would greatly enhance the manuscript’s clarity and depth.

---

### Official Review · Reviewer_dLWo · 2024-10-25

**Soundness:** 2
**Presentation:** 3
**Contribution:** 3
**Rating:** 5
**Confidence:** 4

**Summary:**

In this paper, the author introduces a new framework utilizing the Residual-Energy score and the Debias estimator to alleviate the performance decline induced by long-tailed distributions in visual classification tasks. The author has conducted extensive experiments under both train-from-scratch and fine-tune paradigms, demonstrating that this method outperforms current state-of-the-art approaches.

**Strengths:**

1. Extensive experiments with comparisons against a variety of baselines have demonstrated the validity of this approach.
2. There are supporting theorems that demonstrate the unbiasedness of the Debias estimator.
3. The paper is clearly written and easy to follow.

**Weaknesses:**

1. It is unclear why the author chooses to use the Residual-energy score instead of the original energy score. The motivation for removing the logits of the target class is not discussed. If the Residual-energy score is indeed superior to the original energy score, an experimental comparison or theoretical analysis is necessary to substantiate this claim.

2. The original energy score includes a temperature parameter T, which is neglected in the Residual-energy score. If setting T = 1 is the best practice, at least a theoretical or empirical comparison is needed to validate this. If not, there may be room for improvement.

3. The paper claims in Section 3.2 that the energy score can reflect more sensitive changes than the softmax score, but it only provides a illustrative figure in Fig. 2, lacking a theoretically robust proof.

4. Hyper-parameter analysis in Appendix A. 4 merely lists a wall of numbers, lacking an analysis of the data. Additionally, the result for ViT-B/32 with 200 Epochs in Appendix Table 2 is missing, and the author does not explain why the table is incomplete.

5. The manuscript appears to have not been carefully proofread, containing various typos, including 'tipycal' at line 269, 'Finnally' at line 348, and 'traning from scratch' at line 373, etc.

W1 and W2 are my primary concerns.

**Questions:**

1. There are other energy-based scores that seem to have a similar effect to the Residual-energy score, such as those proposed in [1-3]. What's the difference between Resudual-energy score and these other scores?

2. The author demonstrates the unbiasedness of the Debias estimator. Is it possible to also discuss its efficency and consistency?

[1] Weitang Liu et al. Energy-based Out-of-distribution Detection. NIPS2020.

[2] Ziqian Lin et al. Mood: Multi-level out-of-distribution detection. CVPR2021.

[3] Ru Peng et al. Energy-based Automated Model Evaluation. ICLR2024.

---

### Official Review · Reviewer_kKWu · 2024-10-30

**Soundness:** 3
**Presentation:** 3
**Contribution:** 2
**Rating:** 5
**Confidence:** 3

**Summary:**

The authors tackle the challenge of ensuring model decisions remain unaffected by training data biases, crucial for safe deployment in real-world scenarios with long-tailed distributions. While existing methods improve overall performance, they often compromise specific class accuracy, limiting long-tailed learning effectiveness.

Through a mathematical analysis, the authors identify issues with Empirical Risk Minimization (ERM): Negative Log-Likelihood (NLL) metrics struggle with class distinction due to softmax reliance, and ERM’s naive estimator is biased toward head classes.

To address these, the authors propose Re-Debias, combining a Residual-Energy score for clearer predictions and a Debias estimator that corrects class biases. Tested on iNaturalist18, ImageNet-LT, and CIFAR10/100-LT benchmarks, Re-Debias outperforms state-of-the-art methods, improving balanced performance across classes.

**Strengths:**

1. The paper is clearly written, well-formatted, and organized.

2. The authors conducted extensive experiments across various datasets and model architectures, comparing numerous baselines, all of which demonstrate the effectiveness of the proposed method.

**Weaknesses:**

1. The paper lacks sufficient innovation, and its motivation is not compelling. In Figure 2, the authors illustrate their approach using only two samples, which is unconvincing.

2. The paper lacks theoretical explanations; while the experiments show that Re-Debias outperforms other methods, there is insufficient clarification on why Re-Debias works effectively.

**Questions:**

1. Could the authors include more samples in the motivation section instead of relying on just two examples for illustration?

2. Could more theoretical analysis and validation experiments be conducted to provide a clearer explanation of the benefits of the Residual-Energy-based individual metric?

**Details Of Ethics Concerns:**

None.

---

### Official Review · Reviewer_qSLs · 2024-11-02

**Soundness:** 4
**Presentation:** 3
**Contribution:** 3
**Rating:** 8
**Confidence:** 3

**Summary:**

The paper proposes a method for training a model such that its decision during the inference are independent to the training data distribution. To do that, the authors proposed a novel loss which is decoupled into two components. The first one is related to energy neural models, and it is used to overcome the limitations, in particular the over-confidence, raised from the combination of Cross Entropy loss and Softmax function, while the latter one aims to reduce the impact of the data unbalancing.

**Strengths:**

The paper is well written, and the flow is good. The core thesis and the problem addressed are both clear, as well as the approach to solve such problems.

The method is clearly exposed, making the paper easy to read and evaluate. In particular, I appreciated the discursive approach use to expose the method, which makes clear the process from the problem to the proposal.

The evaluation section contains all the experiments useful to understand if the proposed method is strong against other baselines, and also ablation studies to understand better the method itself.

**Weaknesses:**

I have not found significative weakness in the paper except for some minor typos and errors. Following some of them:

- typo around line 250 (A highlighted)
- I guess Eq 9 contains a typo in the summation indexes (j \ne 1)
- typo around line 348 (Finnally)

**Questions:**

1. In the paper, it is stated that incorporating the new energy function into the NLL loss is suboptimal. Can you support this claim?
2. In the final loss, the unbalancing is mitigated by using log(C·πj) in the loss. However, it is not clear how you estimated those values. In the easiest case you could just count classes ratio of the training set, however, how the loss behaves when the scaling value is not reflected in the test set, or when we are dealing with real-world problems?
3. it is not clear what \overline(y) represents in the residual energy score function
4. How do you implement the k components f used in the Eq 10?

---

### Official Review · Reviewer_qxaX · 2024-11-05

**Soundness:** 3
**Presentation:** 2
**Contribution:** 3
**Rating:** 5
**Confidence:** 5

**Summary:**

This paper proposes an approach for addressing the long-tailed problem. This approach is different from the ones pursued by the current methods that focus on improving individual prediction quality or enhancing aggregate evaluation. Although these methods can improve the overall performance of a model, they often sacrifice performance in some classes, undermining the goals of long-tailed learning.

To address the long-tail problem more holistically, the authors conduct a mathematical analysis of Empirical Risk Minimization (ERM), a widely used model training objective, in long-tailed learning, examining both its individual performance and aggregate evaluation. Their analysis shows that ERM can lead to poor distinction and ambiguity between the training classes, especially when used in combination with NLL. This is because NLL relies heavily on the softmax distribution which introduces biases in the model predictions. Softmax often struggles to distinguish correct from incorrect predictions when their predicted probabilities for the target class are similar.

To "recalibrate" ERM, or reduce the misalignment between softmax-based scores and the input probability density, the authors propose ReDebias, a framework that combines the Residual-Energy score and a Debias estimator. According to the authors, the Residual-Energy score provides a sensitive reflection of prediction quality than softmax-based scores, which enhances prediction precision and reducing ambiguity. The Debias estimator applies causal inference techniques to ensure unbiased estimates during the averaging process, correcting for class-wise biases inherent in the naive estimator.

**Strengths:**

Overall, I find the proposed method pretty interesting. Class imbalance (and long-tailed class distributions) pose a real challenge to deep neural networks, especially when they are trained with ERM. The idea this paper proposes is quite novel and addresses the often overlooked shortcomings of the existing methods. Moreover, I also found the experiments to be solid and the evaluation results quite compelling. Indeed, through extensive validation on long-tailed benchmarks, they have consistently outperformed current baselines, and by a big margin after only a few training epochs. This improved efficiency is highly desirable, especially when training complex networks.

**Weaknesses:**

The main weaknesses of this paper are editorial. Unfortunately, there are many typos in the manuscript, making it at times difficult to read and to thoroughly understand the contributions of the paper. These editorial issues lead to confusing statements. For instance, it is not clear to me whether the adjusted logits $g_y(x)$ should be $g_y(x) = f_y(x) + \log (C\cdot \pi_y)$ or $g_y(x) = f_y(x) - \log (C\cdot \pi_y)$. This is because in Equation 6, the $\arg\max$ is taken over $\exp (f_y(x) - \log (C\cdot \pi_y))$. The authors also claim that $C\cdot n$ is often unknown in the real world, yet the closed form for $\pi_s = \frac{n_c}{|\mathcal{O}|}$, the label frequency, is given and used throughout the manuscript. Finally, it is not clear to me why the residual energy score excludes **always** the first class. At least, from the definition given in Equation 9, $E(x, \bar{y}) = -\log \sum_{y\neq 1}^C e^{f_y(x)}$, this seems to be the case. Should $E(x, \bar{y})$ accounts for all the classes except the class of $y$ or what?

**Questions:**

It would really help if the clarity of the manuscript is improved. This would make their contributions stand out more, and help the reader provide more in-depth feedback to improve the manuscript from a theoretical standpoint. I also suggest the authors to revisit the maths to ensure the correctness of the objectives. Some claims appear often sudden, for instance the approximation $p(y_i|x_i; \theta) \propto e^{f_{y_i}}(x_i)$, could be explained a bit more to the reader.

---

### Note · Authors · 2024-11-13

I have read and agree with the venue's withdrawal policy on behalf of myself and my co-authors.